# Nutritional Intervention Improves Nutrition Outcomes in Stomach and Colon Cancer Patients Receiving Chemotherapy: Finding from a Quasi-Experiment in Vietnam

**DOI:** 10.3390/healthcare9070843

**Published:** 2021-07-04

**Authors:** Le Thi Huong, Duong Thi Phuong, Dang Kim Anh, Phung Lam Toi, Nguyen Le Tuan Anh, Trinh Le Huy, Nguyen Thuy Linh

**Affiliations:** 1Nutrition and Dietetics Department, Hanoi Medical University Hospital, Hanoi 10000, Vietnam; lethihuong@hmu.edu.vn (L.T.H.); linhngthuy@hmu.edu.vn (N.T.L.); 2Institute of Preventive Medicine and Public Health, Hanoi Medical University, Hanoi 10000, Vietnam; kimdanghmu@gmail.com (D.K.A.); dennguyenle@gmail.com (N.L.T.A.); 3Health Strategy and Policy Institute, Ministry of Health, Hanoi 10000, Vietnam; phunglamtoi@hspi.org.vn; 4Oncology Department, Hanoi Medical University, Hanoi 10000, Vietnam; trinhlehuy@hmu.edu.vn

**Keywords:** nutritional intervention, cancer, chemotherapy, outcomes, Vietnam

## Abstract

Background: Evidence on the effects of nutritional interventions on gastrointestinal cancer patients receiving chemotherapy is not well documented. This study aims to assess the effects of nutritional intervention in patients diagnosed with stomach and colon cancer receiving chemotherapy in Vietnam. Methods: A quasi-experiment with intervention and control groups for pre- and post-intervention was carried out in cancer patients receiving chemotherapy in a university hospital in Vietnam. Patients in the intervention group were provided nutritional counseling, personalized specific dietary advice, and received oral nutrition supplements (ONSs) while patients in the control group only received nutrition counseling. Results: The weight in the intervention and control group after 2 months increased significantly by 1.4 ± 2.6 kg and 0.4 ± 2.3 kg, respectively. Muscle mass increased by 1.2 ± 4.1 cm in the intervention group, while those in the control group decreased by 0.55 ± 2.77 cm. There was no statistical significance between two groups after intervention in terms of Mid–Upper Arm Circumference (MUAC) and percentage of fat. The percentage of malnutrition based on the Scored Patient-Generated Subjective Global Assessment (PG-SGA) and Body Mass Index (BMI) declined after the intervention in both groups. According to the average treatment effect on the treated (ATT) using the propensity score matching and DiD method, participants receiving the intervention were more likely to have a higher score of weight (Coef = 0.84; 95%CI = 0.47; 2.16) and muscle mass (Coef = 1.08; 95%CI = 0.09; 2.06) between pre- and post-intervention. By contrast, the PG-SGA scores on treated participants were more likely to decrease after the intervention (Coef = −1.28; 95%CI = −4.39; −0.84). After matching, being female, living in rural areas, or having stomach cancer were still positively related to being moderately/severely malnourished by the PG-SGA, and these findings were statistically significant (*p* < 0.05). Conclusion: The nutritional interventions had a positive effect on weight gain, muscle mass, and reduced malnutrition. Further studies with a longer follow-up duration are needed to confirm the effects of the intervention.

## 1. Introduction

Cancer is a complex disease that results from multiple interactions between genes and the environment, and is regarded as one of the current leading causes of mortality worldwide [1]. Malnutrition, sarcopenia and cachexia among cancer patients, especially those with gastrointestinal cancers, are caused by several multifactorial disorders and can influence the survival and recovery of cancer patients [2,3,4]. Malnutrition from loss of appetite, indigestion, malabsorption, and metabolic problems is a common condition in cancer patients undergoing treatment [5]. Additionally, cancer patients receiving chemotherapy are also affected by side effects of the treatment such as appetite loss, dyspepsia, fatigue, constipation, diarrhea, dysphagia, changes in sensitivity to food temperature, xerostomia, anemia, and early satiation [6]. These factors are associated with decreased food intake, decreased nutritional absorption, changes in body composition, and can consequently cause malnutrition and cachexia [2,7].

Malnutrition in oncology patients often involves cancer cachexia, which is characterized by progressive muscle wasting that cannot be completely reversed by conventional nutrition support [8]. Studies have demonstrated that weight loss in patients with cancer is all independently associated with an unfavorable prognosis and increased toxicity of anticancer treatments, resulting in reductions in or interruptions of scheduled treatment [5]. The goal of nutritional intervention for such patients [9] is to improve their nutritional condition by mitigating treatment side effects, offering individualized patient care, and improving food intake while respecting patients’ food habits with regard to administering more progressive interventions. Improved nutritional condition will enhance both treatment and quality of life in cancer patients [10,11].

In patients with gastrointestinal cancer such as stomach and colon cancer, studies suggested the benefits of nutritional interventions [12,13,14]. However, few studies focus on the effects of nutritional intervention on patients with cancer in the digestive system who receive chemotherapy [15,16]. Besides, nutritional recommendations for this group also have not been harmonized and specified for settings and races [12]. In this study, we apply nutrition recommendations for cancer patients from organizations and nutritional experts into the culinary culture context, taking into consideration the habits and preferences of Vietnamese people, thereby aiming to supply and improve the food intake of cancer patients.

Vietnam is a developing country with a high burden of cancer. In the country, the prevalence of gastric cancer was 10.6% in 2018, and more than 7000 deaths were attributed to this cancer [17]. The proportion of malnutrition based on the Scored Patient-Generated Subjective Global Assessment (PG-SGA) among gastrointestinal cancer patients in Vietnam ranges from 46.9% to 59.3% [10,11]. Since the nutritional status plays a critical role in the prognosis of treatment for cancer patients [7,18], the provision of nutritional interventions in association with clinical practices for gastrointestinal cancer patients is essential, especially those in receipt of chemotherapy. To date, there have been limited studies investigating the clinical effects of nutritional interventions on gastrointestinal cancer patients receiving chemotherapy in Vietnam. Therefore, this study aimed to assess the clinical effects of nutritional intervention in patients diagnosed with stomach and colon cancer and treated by chemotherapy. Not only do the results of this study provide evidence on the effects of nutritional intervention for cancer patients, but it also recommends the integration of personalized nutrition support for cancer patients in clinical practice in Vietnam.

## 2. Materials and Methods

### 2.1. Study Design

This study was a quasi-experimental research with intervention and control groups for pre- and post-intervention assessment (study registration: ClinicalTrials.gov [NCT04517708] (accessed on 3 July 2021)). A convenience sampling technique was used for recruiting participants from July 2017 to July 2019 at Hanoi Medical University Hospital (HMUH).

### 2.2. Participants

Adult patients (i.e., aged ≥ 18 years) with histologically verified stomach or colon cancer, eligible for chemotherapy, and indicated for oral feeding were considered for study participation. Participants in both groups were paired together according to the following criteria: (1) age group: <40 years old, 40–65 years old, and >65 years old; (2) gender: male and female; (3) types of cancer: stomach cancer and colon cancer; (4) disease stages: stage 1–2 and stage 3–4. Patients were excluded from the study if they: (1) had contraindications to enteral nutrition, (2) underwent terminal palliative care, (3) had other chronic comorbidities such as diabetes, renal and/or heart failure.

In this study, we utilized sample size calculation when the outcome measure was a continuous variable [19] with δ = 10% (difference of 10% is widely accepted as clinically significant [19]), s = 13.3, the β = 0.2 and α = 0.05. Therefore, the sample size in each group was 30 patients. In fact, the final sample size with a refusal rate of 20% was 60 patients for each group at the baseline.

After screening for eligibility criteria, a total of 120 patients were initially invited to participate in this study, in which patients were allocated in the intervention group if they agreed to follow the personalized diet. None of invited patients refused to participate in the study. However, after three weeks, one and three patients in the intervention and control group, respectively, withdrew from study. At the end of follow up (8 weeks), the total number of participants in the intervention and control groups were 53 and 50, respectively (Figure 1).

### 2.3. Developing the Intervention Protocol

Step 1: The literature review was conducted regarding methods and tools which have been primarily used for screening, assessing nutritional status, and the process of nutritional intervention. The results of the literature review were compared with current practice at HMU hospital.

Step 2: An intervention protocol was developed.

Step 3: Consultations with approximately 30 nutritional experts took place to finetune the intervention protocol.

Step 4: We developed the tools, documents, and materials related to the intervention, including:Consultancy leaflets.Menus based on energy levels following the demand recommendations by the European Society for Clinical Nutrition and Metabolism (ESPEN), with total energy being 30 kcal/kg of body weight/day, protein from 1.2 to 1.6 g/kg of body weight/day and 2 g EPA/day [5]. Depending on the weight and digestive condition of each patient, we classified the menus into different energy levels, which consisted of 1500–1600 kcal/day, 1700–1800 kcal/day, or 1900–2000 kcal/day. Patients were recommended to eat small and frequent meals 5–6 times a day to increase their overall intake. If the supplements were feasible and sensible, patients consumed two snacks with 200–400 mL ONS/day (Appendix A).Menus and high-energy soups were prepared, and these menus were evaluated based on suitability and acceptance of the patients.A handbook was prepared for processing high-energy soup preparations from common foods for cancer patients.

Step 5. The protocol was applied to the intervention study (Figure 2).


Nutritional screening according to the MST (malnutrition screening tool) by dietitians within the first 24 h after the hospital admission took place at two time points (baseline—T0 and 2 months after chemotherapy—T1).Nutritional status was assessed according to anthropometric measurements (weight, BMI, percentage of weight loss, MUAC, muscle mass, fat mass); the PG-SGA; albumin, prealbumin and total protein at two time points (baseline—T0 and 2 months after chemotherapy—T1).Nutritional diagnosis.The control group received nutritional status screening and assessment, and nutritional counselling. They could freely select their own diet. In contrast, patients in the intervention group were treated with the intervention regimen, which consisted of: (1) nutritional counseling; (2) hospital meals: each patient was assigned a specific menu prepared by research members during the time of staying at the hospital; (3) instruction on food preparation at home: before discharge, patients were instructed on preparing their diet at home with high energy and protein intake as recommended and given formula milk (6 teaspoons, equivalent to 54 g of powdered milk with 180 mL warm water per time) within two months [20].To monitor the intervention process, the dieticians continuously evaluated the daily diet of each patient during hospitalization and tele-counseling every two weeks for weight monitoring and nutritional support advice after discharge. Patients were re-evaluated in terms of their nutritional status at every hospital admission for chemotherapy.The target of the nutritional intervention was to ensure the ability to meet the recommended energy and protein of patients. The criteria for making adjustments to the intervention was practicable and implemented by the patient. In addition, an appropriate treatment tailored to the patients’ eating preferences and digestive and absorptive capacity was taken into consideration.


### 2.4. Nutritional Evaluation

We evaluated the anthropometry of participants, including height and weight, Mid–Upper Arm Circumference (MUAC), subcutaneous fat thickness, fat mass, and muscle mass. (1) We measured standing height using a standard height ruler with an accuracy of 0.1 cm. For weight measurement, we used an electronic scale with an accuracy of 0.1 kg, where the calibration was carried out before measurement. Both height and weight measurements were performed twice, and the average value was recorded. (2) MUAC was measured at the mid-point between the tip of the shoulder and the tip of the elbow in the left upper arm. (3) The thickness of the subcutaneous fat was measured by Harpenden’s skin calipers at the triceps muscle (i.e., triceps fat thickness (TFT). (4) Muscle mass covered the measurement of the smooth and skeletal muscles as well as water in the body. Bone mass is the overall bone mineral density measurement of the body. (5) This study applied bioelectrical impedance analysis (BIA) to assess body composition (muscle mass, fat mass) by using the Tanita BC-758 scale (Tanita Corp., Tokyo, Japan). The Scored Patient-Generated Subjective Global Assessment (PG-SGA) was used as the preeminent interdisciplinary patient assessment among patients with oncology [21]. This tool includes (a) four patient-generated historical components (Weight History, Food Intake, Symptoms, Activities, and Function); (b) the professional part (Diagnosis, Metabolic stress, and Physical Exam); (c) the Global Assessment (A = well-nourished, B = moderately malnourished, C = severely malnourished); (d) the total score, and nutritional triage recommendations. Moreover, the blood albumin and total serum protein were tested. In this study, malnutrition was determined if (Body Mass Index) BMI ≤ 18.5 kg/m^2^ [22], albumin < 35g/L, or total serum protein < 65 g/L [23]. The measurements were taken in the first 24 h after admission and 2 months after following up (Figure 1).

### 2.5. Data Analysis

Data analysis was performed by STATA 15.1 (College Station, TX: LLC). Descriptive statistics were used to analyze the patient demographics. Categorical data were analyzed by the χ^2^ test, Fisher’s exact test, and McNemar’s Chi-square test where applicable. McNemar’s Chi-square test was used to test proportional changes in nutritional status between before (T0) and after the intervention (T1) in different groups of BMI and PG-SGA, whereas continuous data were checked for normality; all data were normally distributed, except for TFT. Accordingly, the paired Student’s *T*-test was applied to test the difference between T0 and after T1 regarding weight, muscle mass, MUAC, and fat mass. Meanwhile, the Wilcoxon sign-rank test was applied for the analysis of the difference in TFT between T0 and T1. A paired Student’s *T*-test was also employed to test the difference between before and after the intervention in each group if the data followed normal distribution. The *p*-value of 0.05 was applied as the level of statistical significance.

In order to control the non-random allocation of participants, we used the propensity score with a probit regression model, which predicted the likelihood of being included in the intervention group. Participants were matched 1:1 into the control group and intervention (treated group). Variables were selected as criteria for matching, including age, gender, occupation, educational level, economic condition, living area, type of cancer, stage of cancer, number of chemotherapy treatments, and having comorbidities. As participants matched only once, no replacement was applied. Figure 3 presents the standardized differences between the intervention and control groups before and after matching. After propensity score matching, the difference in differences method (DiD) was utilized to compare the changes over time of nutritional indicators in the control group and treated group and estimate the effect of the intervention. At first, this method provided unbiased effect estimates if the trend of change was similar between treated and control groups without the intervention. Then, the average increase in the control group was eliminated from the average gain in the treated group. The realistic effect of the intervention was measured by the average treatment on treated (ATT), which was the expected causal effect of the treatment for individuals in the treatment group.

## 3. Results

### 3.1. Participants’ Characteristics

Table 1 depicts the characteristics of participants before and after the propensity score matching. A total of 50 patients of the control group and 53 patients of the intervention group were identified. Before matching, the mean age of participants was 54.9 ± 10.6 years and 58.2 ± 10.0 years in the intervention and control group, respectively. Nearly two-thirds of the patients were male (62.3% and 62.0%), and the majority of the patients were at stage III and IV of cancer (84.9% and 82.0%). The percentage of patients with comorbidities in the control group was higher than that of the intervention group (28.0% and 7.5%, respectively, *p* < 0.05). After 1:1 ratio propensity score matching, 31 patients in each group were retained for comparison. No significant differences in previously associated covariates were determined between the two groups.

### 3.2. Effectiveness of Nutritional Intervention

The nutritional status of participants in both groups were assessed and presented in Table 2. The mean of weight and muscle mass in the intervention group significantly increased after the intervention. The percentage of participants being moderately/severely malnourished or having BMI ≤ 18.5 kg/m^2^ of both groups in T1 were significantly lower than T0. Regarding the intervention group, the PG-SGA scores were significantly reduced, and the BMI scores were significantly increased post-intervention.

Considering the types and stages of diseases, the mean weight gain was statistically significant among colon cancer (*p* = 0.007). Particularly, after the intervention, the average weight gain among colon cancer patients was 2.5 ± 1.8 kg in the intervention group, whereas this figure for the control group was 0.9 ± 2.2 kg. There was no statistical significance between the intervention and control groups among stomach cancer patients (*p* = 0.4), disease stage I–II (*p* = 0.3), and disease stage III–IV (*p* = 0.1). Regarding muscle mass, even though there were statistically significant differences after intervention among stomach cancer (*p* = 0.04) and stage III–IV (*p* = 0.04) in the intervention group, there was no statistical significance between the intervention and control group (*p* > 0.05) (Table 3).

The MUAC in colon cancer patients of the intervention group significantly increased (0.8 ± 0.89 cm), (*p* = 0.0001), whereas the control group decreased 0.54 ± 3.5 cm (*p* = 0.42). There was a statistically significant difference between the intervention and control group of colon cancer patients regarding the MUAC (*p* = 0.002). Similarly, there was also a statistically significant difference in the MUAC between intervention and control group among the patients’ stage III–IV (*p* = 0.006). Meanwhile, there was no difference between the two groups in terms of the percentage of fat and TFT by diseases and stage (Table 3).

Similarly, there was also a statistically significant difference in PG-SGA scores and albumin levels between the intervention and control group among patients at stage III–IV and both types of cancer (*p* < 0.05) (Table 3).

Table 4 shows the results of the average treatment effect on the treated (ATT) using the propensity score matching and DiD method. Based on the findings, the intervention had an effective impact on the nutritional status of cancer patients. Participants receiving the intervention were more likely to have a higher score of weight (Coef = 0.84; 95%CI = 0.47; 2.16) and muscle mass (Coef = 1.08; 95%CI = 0.09; 2.06) between pre- and post-intervention. By contrast, the PG–SGA scores in treated participants were more likely to decrease after the intervention (Coef = −1.28; 95%CI = −4.39; −0.84).

Factors associated with being moderately/severely malnourished by the PG-SGA are shown in Table 5. Regression analyses revealed that females were more likely to be moderately/severely malnourished than males (OR = 1.70; 95%CI = 1.38; 7.66). In addition, living in urban areas or having colon cancer were positively associated with being well-nourished compared to those living in rural areas or having stomach cancer (OR = 0.82; 95%CI = 0.31; 0.91 and OR = 0.32; 95%CI = 0.11; 0.92, respectively). After matching, being female, living in rural areas, or having stomach cancer were still positively related to being moderately/severely malnourished, and these findings were statistically significant.

## 4. Discussion

In Vietnam, the field of clinical nutrition has really developed in recent years. The benefit of nutritional support has not been adequately considered by oncologists. Similar to previous studies, some factors acting as barriers to nutritional care included a lack of clear guidelines, a lack of knowledge or training in this area, time constraints preventing referral for, or direct nutritional interventions, and a lack of time to carry out implementation processes for nutritional care (assessment, intervention, and follow up) [24,25]. To our knowledge, this study was one of the scarcity of studies in Vietnam, which investigates the effects of nutritional intervention on the nutritional status of patients with gastrointestinal cancer. Findings from this study provide valuable evidence for clinicians, especially oncologists and nutritionists, who would benefit from nutritional support and nutrition intervention processes for cancer patients in the context of Vietnam. The development of standardized nutritional care pathways is complex and requires nutritional care to be fully integrated as part of the multimodal care process in cancer patients. Our study not only provides evidence to investigate the effects of a nutritional intervention protocol for gastrointestinal cancer patients receiving chemotherapy, but also to build the menus and high-energy soup preparations according to the Vietnamese Food Composition Table, compatible with the taste and culinary culture of Vietnamese people from common and popular foods to help patients increase their dietary intake, thereby helping to improve their nutritional status.

The primary outcomes analyzed in this study are changed in nutritional status outcomes before and after nutritional intervention and comparison between the treatment and control group. Our study showed that nutritional intervention with dietitian counseling, particularly an individualized specific diet, could significantly improve the anthropometry and reduce the malnutrition of cancer patients. This is consistent with the guideline of ESPEN, stating that “Nutrition counseling by a health care professional is regarded as the 1^st^ line of nutrition therapy. Professional counseling is a dedicated and repeated professional communication process that aims to provide patients with a thorough understanding of nutritional topics that can lead to lasting changes in eating habits. Clearly, the best way to maintain or increase energy and protein intake is with normal food” [5].

Our data showed that the increase in body weight was statistically significant in the intervention group compared to the control group. Even though this was a modest increase in comparison with the previous report [26] due to our short following up duration (2 months), the results were still optimistic, as weight gain signified good nutritional status [27]. We also found the effect of the intervention on gained muscle mass rather than the percentage of fat mass, indicating that the main contribution to the gained weight was muscle rather than fat. The muscle mass reduction was more likely to cause surgical complications and treatment-induced toxicity during systemic anticancer therapy, whereas greater muscle mass was associated with the improved survival of cancer patients [5,28]. Therefore, the nutritional interventions must place a considerable emphasis on the maintenance or improvement of patients’ muscle mass. Since the physical activity and performance status of patients are mostly impaired by cancer and this is often accompanied by a further loss of muscle mass, a combination of nutritional and physical therapy is recommended [5,29]. Previous evidence suggested the benefits of a combination of nutrition and physical activity intervention on cancer patients [30,31,32]. It is noteworthy that most of these studies focus on cancer survivors [31] or palliative care [32]. Future research, therefore, might be interested in investigating the benefits of a combination of nutritional and physical activity intervention on patients during chemotherapy.

Interestingly, we found that the muscle mass increased significantly in the group of stomach cancer (*p* = 0.04) and those who at stage III–IV (*p* = 0.04). Our finding is consistent with the study by Silvers et al., reporting that nutritional intervention attenuates the loss of body weight compared to standard care among upper gastrointestinal cancers [33].

There was a statistically significant difference between the two groups of colon cancer patients regarding the MUAC, as this measurement increased in the intervention group and decreased in the control group two months after the chemotherapy. Our finding was in accordance with the previous meta-analysis by Baldwin et al., which showed that there was a difference in the MUAC, favoring participants who received dietary advice, with a pooled mean difference of 0.81 cm (95% CI 0.31 to 1.31) (*p* = 0.001) [26].

Moreover, our data suggested that the interventions had a considerable beneficial effect on the nutritional status of the patients. The baseline malnutrition rate in both groups was substantially high, at approximately 80% according to the PG-SGA, which was consistent with previous studies [33,34,35]. Meanwhile, the BMI results revealed that the malnutrition rates at baseline in both groups were only over 30%, which was much lower than that of the PG-SGA. This strengthened the previous statement that the PG-SGA was a multidimensional tool to assess malnutrition and more likely to yield more malnourished cases than BMI [15]. Therefore, the selection of an appropriate tool for assessing nutrition status should be carefully considered. Besides, the improvements in different methods of nutritional status assessment in the intervention group in this study were similar to previous studies that investigated the effect of nutritional counseling (with no ONS prescription) on nutritional status and reported positive effects of nutritional counseling on different methods of nutritional status, including PG-SGA scores [36,37] and BMI [36].

Several studies showed that a normal serum albumin level during chemotherapy was associated with the positive outcomes and reduced side effects of anticancer treatments [38]. In our study, we found that the rates of albumin deficiency (<35 g/L) and total serum protein deficiency (<65 g/L) increased significantly in the control group, and the mean albumin and total serum protein level in the control group were lower than the intervention group, especially for the colon cancer group and stage III–IV, suggesting that patients in the control group might have a worse prognosis in this study than those in the intervention group.

In addition, factors associated with being moderately/severely malnourished by the PG-SGA after matching and regression analyses revealed showed that being female, living in rural areas, or having stomach cancer were still positively related to being moderately/severely malnourished, and these findings were statistically significant. Our finding is consistent with the reports of Liyan Zhang [34] and Du YP [39]. It has been indicated that gastric cancer patients had a higher risk of malnutrition compared to colon cancer. The study by Hebuterne X (2014) [40] on 1545 cancer patients showed that the prevalence of malnutrition according to BMI by disease site was as follows: colon/rectum (39.3%); head and neck (48.9%) and esophagus and/or stomach cancer (60.2%). The main causes of malnutrition in patients with upper gastrointestinal tract cancer are the abnormal increased metabolism included by the tumor, difficulty in eating and digestion, and side effects of anticancer therapy such as nausea, vomiting, fatigue and pain [5,34]. Additionally, the economic status and availability of food should be considered in the planning process of nutritional intervention.

Several limitations of our study warrant consideration. First, our follow-up duration was quite short (i.e., 2 months) compared to other studies, making it difficult to extrapolate the long-term outcomes. However, our study demonstrated that an early individualized nutritional intervention during chemotherapy was feasible to improve the dietary intake as well as the nutritional status of the patients. The participants’ selection for our study was also limited as it was not double-blinded, but the collection and evaluation of data were strictly conducted by a certified dietitian to minimize any observer’s biases. Despite such factors, we observed significant improvement in weight, muscle mass, as well as PG-SGA scores and albumin levels, and total serum protein in the intervention group compared to the control group, suggesting there are advantages to using nutritional counseling and interventions in patients receiving chemotherapy.

In summary, nutrition intervention with counseling by a dietitian and the prescription of a protein- and energy-dense ONS enriched with EPA, together with chemotherapy, resulted in significant improvements in nutritional status and clinically significant improvements in weight and lean body mass in patients receiving chemotherapy over 2 months. Based on the results of this study, the nutritional and educational interventions (i.e., instruction for food preparation) are worthy of clinical application. While the guidelines in nutritional support for cancer patients are widely applied in many countries, it is still limited in Vietnam [35]. Our findings suggested that nutrition intervention should be supplemented to the standard of care for oncology patients nationwide.

## 5. Conclusions

In conclusion, nutrition intervention with counseling by a dietitian combined with the prescription of specific diet improved weight, muscle mass, and nutritional status by the PG-SGA in cancer patients receiving chemotherapy. Further research for a long time and specific types of cancer in the form of a randomized controlled trial is required to confirm these results in cancer patients receiving chemotherapy.

## Figures and Tables

**Figure 1 healthcare-09-00843-f001:**
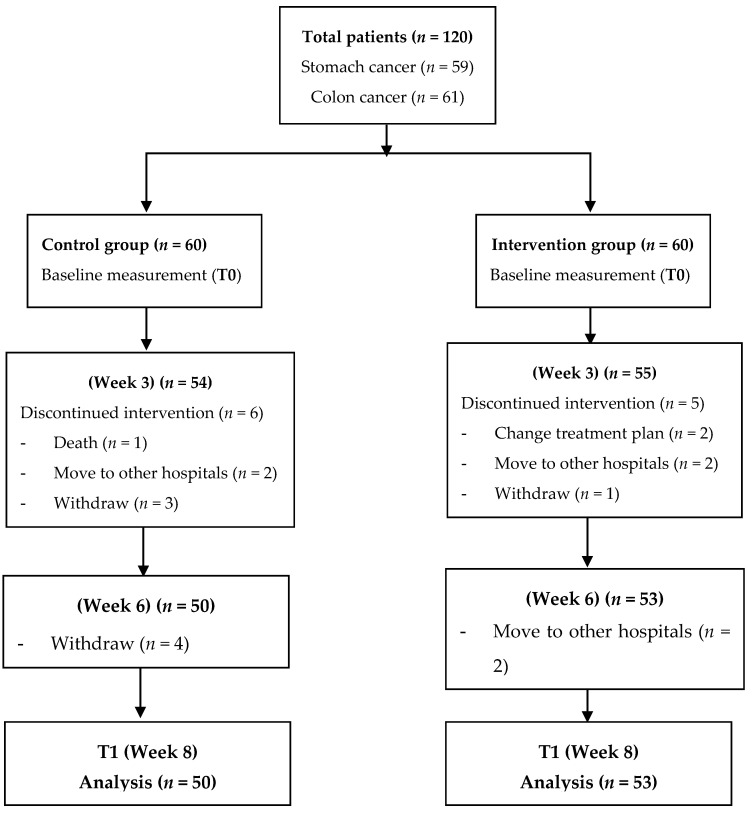
Flow diagram of study design.

**Figure 2 healthcare-09-00843-f002:**
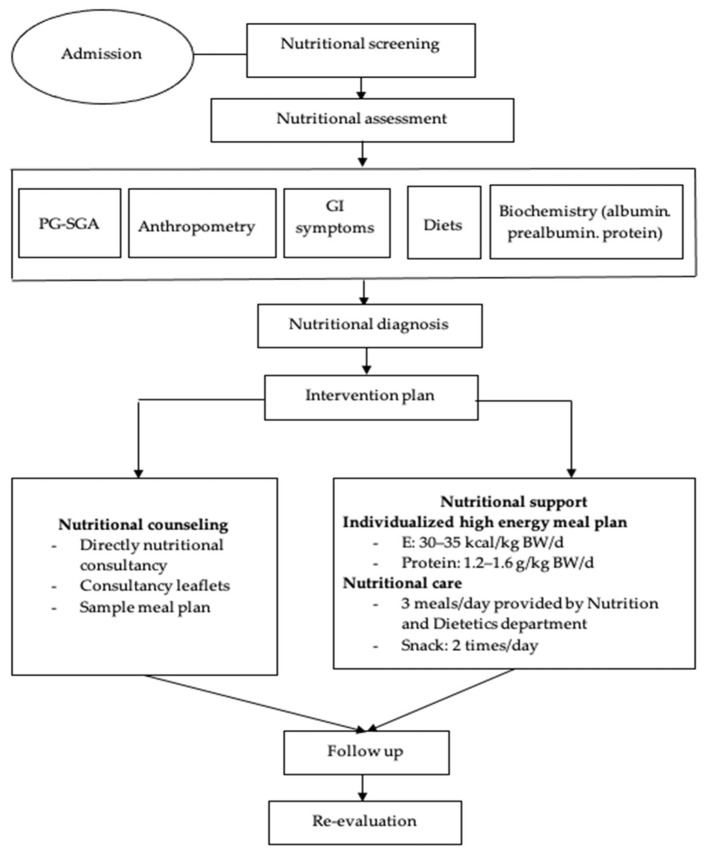
The process diagram of nutritional intervention for intervention group.

**Figure 3 healthcare-09-00843-f003:**
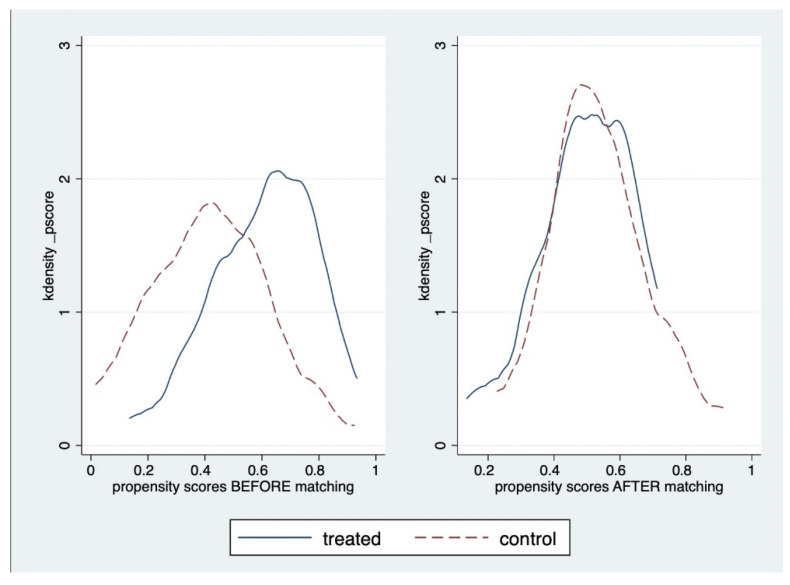
Propensity score distribution between intervention and control groups before and after matching.

**Table 1 healthcare-09-00843-t001:** Characteristics of participants before and after propensity score matching.

Characteristics of Participants	Before Propensity Score Matching					After Propensity Score Matching (1:1)				
	Control Group (*n* = 50)		Intervention Group (*n* = 53)		*p*-Value	Control Group (*n* = 31)		Intervention Group (*n* = 31)		*p*-Value
	*n*	%	*n*	%		*n*	%	*n*	%	
**Gender**										
Males	31	62.0	33	62.3	0.98 ^a^	22	71.0	22	71.0	1.00 ^a^
Females	19	38.0	20	37.7		9	29.0	9	29.0	
**Educational level**										
Under high school	25	50.0	25	47.2	0.60 ^a^	14	45.2	14	45.2	1.00 ^a^
Highschool	13	26.0	10	18.9		6	19.3	6	19.3	
Intermediate college	4	8.0	8	15.0		4	12.9	4	12.9	
University/postgraduate	8	16.0	10	18.9		7	22.6	7	22.6	
**Occupation**										
Office workers	4	8.0	7	13.2	0.60 ^a^	4	12.9	4	12.9	0.76 ^a^
Farmers/workers	12	24.0	14	26.4		5	16.1	8	25.8	
Retired	22	44.0	18	34.0		13	41.9	10	32.3	
Doing household	12	24.0	14	26.4		9	29.1	9	29.1	
**Economic conditions**										
Poor/near-poor households	2	4.0	2	3.8	0.14 ^b^	2	6.4	1	3.2	0.22 ^b^
Normal/rich	48	96.0	51	96.2		29	93.6	30	96.8	
**Living area**										
Rural	25	50.0	27	51.0	0.92 ^a^	15	48.4	14	45.2	0.80 ^a^
Urban	25	50.0	26	49.0		16	51.6	17	54.8	
**Type of disease**										
Stomach cancer	22	44.0	29	54.7	0.28 ^a^	15	48.4	15	48.4	1.00 ^a^
Colon cancer	28	56.0	24	45.3		16	51.6	16	51.6	
**Stage of cancer**										
I–II	9	18.0	8	15.1	0.69 ^a^	4	12.9	8	25.8	0.26 ^b^
III–IV	41	82.0	45	84.9		27	87.1	23	74.2	
**Number of chemotherapy treatments**										
0	19	38.0	21	39.6	0.87 ^a^	13	41.9	13	41.9	1.00 ^a^
1	31	62.0	32	60.4		18	58.1	18	58.1	
**Having comorbidities**										
No	36	72.0	49	92.5	0.02 ^a^	26	83.9	27	87.1	0.30 ^a^
Yes	14	28.0	4	7.5		5	16.1	4	12.9	
	**Mean**	**SD**	**Mean**	**SD**	***p*-Value**	**Mean**	**SD**	**Mean**	**SD**	***p*-Value**
Age	58.2	10.0	54.9	10.6	0.11 ^c^	55.2	10.2	57.3	8.9	0.58 ^c^

^a^: Chi-squared test; ^b^: Fisher’s exact test; ^c^: Mann–Whitney test.

**Table 2 healthcare-09-00843-t002:** Nutritional status of participants before and after intervention of control and treated group.

	Control Group (*n* = 50)		Intervention Group (*n* = 53)		*p*-Value (1–2)	*p*-Value (3–4)
	T0 (1)	T1 (2)	T0 (3)	T1 (4)		
**Weight**	50.5 ± 7.6	50.9 ±7.1	50.2 ± 7.4	51.6 ± 7.8	0.19 ^¶^	<0.01 ^¶^
**Muscle mass**	37.0 ±5.7	37.6 ±5.6	36.5 ± 5.8	37.7 ±6.6	0.16^¶^	0.02 ^¶^
**MUAC**	25.2 ±3.1	24.6 ±3.1	25.3 ± 2.5	25.6 ±2.9	0.16 ^¶^	0.29 ^¶^
**Fat mass (%)**	23.2 ± 6.5	21.5 ± 6.6	23.0 ± 7.5	22.6 ± 7.5	0.02 ^¶^	0.30 ^¶^
**Albumin (g/L)**	35.8 ± 7.2	30.7 ± 11.5	39.2 ± 5.8	36.7 ± 5.5	0.01 *	0.05 *
**Protein (g/L)**	62.6 ± 13.6	52.1 ± 15.9	58.8 ± 10.6	57.8 ± 8.5	0.02 *	0.57 *
**PG-SGA score**	11.7 ± 5.1	10.9 ± 6.2	13.1 ± 5.6	8.9 ± 6.0	0.39 *	<0.01 *
**PG-SGA classification**						
PG-SGA A (Well-nourished)	9 (18.0)	20 (40.0)	12 (22.6)	34 (64.2)	<0.01 *^#^*	<0.01 *^#^*
PG-SGA B and C (moderately/severely malnourished)	41 (82.0)	30 (60.0)	41 (77.4)	19 (35.8)		
**BMI score**	19.6 ± 2.4	19.8 ± 2.2	19.7 ± 2.2	20.3 ± 2.4	0.12 ^¶^	<0.01 ^¶^
**BMI classification**						
≤18.5 kg/m^2^	20 (40.0)	13 (26.0)	17 (32.1)	10 (18.9)	0.04 *^#^*	0.02 *^#^*
>18.5 kg/m^2^	30 (60.0)	37 (74.0)	36 (67.9)	43 (81.1)		

^#^: McNemar’s Chi-squared test; ^¶^: Paired–samples T test; *: Wilcoxon signed-rank test.

**Table 3 healthcare-09-00843-t003:** Changes in body index and nutritional status by type of cancers and stages.

	Intervention (*n* = 53) (Mean ± SD)			Control (*n* = 50) (Mean ± SD)			P _I-C_
	T0	T1	P_1_	T0	T1	P_2_	
**Weight (kg)**							
Stomach cancer	49.5 ± 8.5	49.9 ± 8.6	0.46	51.0 ± 7.8	50.8 ± 6.4	0.66	0.4
Colon cancer	50.99 ± 6.1	53.5 ± 6.4	<0.0001	50.1 ± 7.6	51.1 ± 7.6	0.02	0.007
Stage I–II	50.2 ± 5.5	52.0 ± 6.2	0.04	47.6 ± 5.9	48.2 ± 5.6	0.55	0.3
Stage III–IV	50.2 ± 7.8	51.5 ± 8.1	0.001	51.2 ± 7.9	51.6 ± 7.3	0.26	0.1
**Muscle mass (kg)**							
Stomach cancer	36.3 ± 5.8	37.9 ± 7.1	0.04	38.2 ± 6.0	38.5 ± 5.7	0.52	0.29
Colon cancer	36.7 ± 5.9	37.4 ± 6.2	0.34	36.1 ± 5.4	36.8 ± 5.5	0.21	0.98
Stage I–II	35.2 ± 4.0	36.4 ± 4.4	0.15	33.8 ± 4.3	34.4 ± 4.4	0.36	0.56
Stage III–IV	36.8 ± 6.0	37.9 ± 7.0	0.04	37.7 ± 5.8	38.2 ± 5.8	0.2	0.14
**MUAC (cm)**							
Stomach cancer	24.99 ± 2.8	24.8 ± 3.2	0.62	25.0 ± 2.5	24.3 ± 2.5	0.19	0.23
Colon cancer	25.7 ± 2.1	26.5 ± 2.1	0.0001	25.3 ± 3.6	24.8 ± 3.5	0.42	0.002
Stage I–II	25.6 ± 2.9	25.8 ± 3.2	0.77	25.3 ± 3.6	25.0 ± 4.3	0.8	0.56
Stage III–IV	25.3 ± 2.5	25.6 ± 2.8	0.3	25.2 ± 3.0	24.5 ± 2.9	0.14	0.006
**Fat mass (%)**							
Stomach cancer	20.96 ± 7.3	20.4 ± 7.8	0.57	21.87 ± 5.5	20.3 ± 6.5	0.13	0.6
Colon cancer	25.5 ± 7.0	25.2 ± 6.4	0.47	24.2 ± 7.1	22.4 ± 6.7	0.02	0.25
Stage I–II	26.2 ± 10.2	25.2 ± 8.9	0.4	24.5 ± 9.6	24.7 ± 8.0	0.8	0.44
Stage III–IV	22.5 ± 6.9	22.2 ± 7.3	0.6	22.9 ± 5.7	20.7 ± 6.2	0.006	0.1
**TFT** **(mm)**							
Stomach cancer	11.9 ± 5.9	15.0 ± 15.2	0.86	10.4 ± 4.9	11.9 ± 4.8	0.45	0.68
Colon cancer	15.4 ± 8.0	15.2 ± 7.2	0.87	11.5 ± 6.9	12.0 ± 5.7	0.52	0.63
Stage I–II	14.7 ± 10.1	16.4 ± 10.6	0.58	11.3 ± 8.4	14.4 ± 7.3	0.3	0.53
Stage III–IV	13.2 ± 6.6	14.9 ± 12.5	0.6	10.9 ± 5.6	11.4 ± 4.7	0.66	0.62
**PG–SGA score**							
Stomach cancer	15.8 ± 5.3	11.0 ± 6.7	0.001	12.9 ± 5.3	12.4 ± 5.5	0.46	0.0013
Colon cancer	10.3 ± 3.9	6.3 ± 3.9	0.0007	11.1 ± 4.5	9.7 ± 6.5	0.33	0.003
Stage I–II	12.3 ± 3.4	9.3 ± 7.8	0.18	12.4 ± 6.8	9.9 ± 7.0	0.26	0.08
Stage III–IV	13.5 ± 5.7	8.8 ± 5.8	0.000	11.8 ± 4.5	11.1 ± 6.0	0.44	0.0001
**Albumin (g/L)**							
Stomach cancer	38.4 ± 5.8	37.1 ± 4.8	0.21	35.0 ± 8.1	29.1 ± 14.1	0.11	0.04
Colon cancer	40.4 ± 6.0	36.1 ± 6.6	0.14	36.4 ± 6.6	32.0 ± 9.1	0.04	0.01
Stage I–II	34.7 ± 7.6	35.8 ± 8.9	0.29	35.9 ± 7.6	30.3 ± 17.1	0.48	0.72
Stage III–IV	39.7 ± 5.6	36.8 ± 5.2	0.04	35.8 ± 7.2	30.8 ± 9.8	0.009	0.001
Total serum Protein (g/L)							
Stomach cancer	59.6 ± 11.2	57.3 ± 6.4	0.46	57.4 ± 13.2	48.6 ± 17.2	0.25	0.17
Colon cancer	59.3 ± 11.7	58.6 ± 10.6	0.94	66.6 ± 13.5	54.6 ± 14.3	0.0045	0.014
Stage I–II	49.9 ± 13.3	54.6 ± 8.0	0.46	63.7 ± 14.4	45.0 ± 17.4	0.25	0.16
Stage III–IV	60.8 ± 10.5	58.2 ± 8.3	0.94	62.3 ± 14.1	53.6 ± 15.0	0.005	0.018

MUAC, Mid–Upper Arm Circumference; TFT, triceps fat thickness. PG-SGA, Patient-Generated Subjective Global Assessment; I, intervention; C: comparator.

**Table 4 healthcare-09-00843-t004:** Average treatment effect on the treated (ATT) using the propensity score matching and DiD method between pre- and post-intervention regarding several nutritional assessment indicators.

	Weight		Muscle Mass		MUAC		Fat Mass		BMI		PG–SGA	
	Coef	95%CI	Coef	95%CI	Coef	95%CI	Coef	95%CI	Coef	95%CI	Coef	95%CI
Average treatment effect on the treated (ATT)	0.84 *	0.47; 2.16	1.08 *	0.09; 2.06	0.1	−0.77; 0.97	1.06	−1.03; 3.14	0.33	–0.17; 1.83	−1.28 *	–4.39; −0.84

* *p* < 0.05; propensity score matching for age, gender, occupation, educational level, economic condition, living area, type of cancer, stage of cancer, number of chemotherapy treatments, having comorbidities.

**Table 5 healthcare-09-00843-t005:** Factors associated with being moderately/severely malnourished, assessed by PG-SGA before and after propensity score matching.

	Before Propensity Score Matching		After Propensity Score Matching (1:1)	
	Moderately/Severely Malnourished by PG–SGA (Yes/No)		Moderately/Severely Malnourished by PG–SGA (Yes/No)	
	OR	95%CI	OR	95%CI
Gender (Female vs. Male)	1.70 *	1.38; 7.66	2.58 *	1.49; 6.51
Age	1.00	0.94; 1.06	1.03	0.98; 1.09
Living area (Urban vs. Rural)	0.82 *	0.31; 0.91	0.91 *	0.43; 0.98
Type of cancer (Colon cancer vs. Stomach cancer)	0.32 *	0.11; 0.92	0.43 *	0.16; 0.95
Weight	0.97	0.88; 1.06	0.99	0.90; 1.09
Bone mass	1.02	0.40; 2.65	1.45	0.58; 3.64
Fat mass	0.94	0.85; 1.03	0.92	0.84; 1.03
MUAC	0.98	0.82; 1.18	0.94	0.75; 1.16

* *p* < 0.05.

## Data Availability

The data presented in this study are available on request from the corresponding author.

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
