# Peer review of "Nutritional Intervention Improves Nutrition Outcomes in Stomach and Colon Cancer Patients Receiving Chemotherapy: Finding from a Quasi-Experiment in Vietnam"

_healthcare, 2021, doi:10.3390/healthcare9070843_

Round 1
Reviewer 1 Report
This is an important study evaluating the effects of nutritional intervention on increasing quality of life in patients with gastrointestinal cancer. However, it needs a major revision. Please see my comments below:
Line 19: unclear statement, personalized specific dietary?
Line 24: abbreviation needs to be defined the first time it appears in the text; MUAC?
Line 33: need to report P-value for statistically significant results.
Line 77-78: it is the wrong statement, please see the links below
https://pubmed.ncbi.nlm.nih.gov/33550719/
https://www.ncbi.nlm.nih.gov/pmc/articles/PMC4855043/
Line 92: The authors chose a wide age range, what is the rationale for the age range?
Line 92: Did participants receive any other therapy or medication? Were they excluded from the study if the condition was metastatic?
Line 100: need to give more details on sample size calculation. Was this calculation based on previous studies?
Line 120: Need to give more detail on ONS and snacks. What were the ingredients, Nutrient contents, brand,...?
Line 128: were participants with malnutrition at the beginning of the study excluded from the study? If they were included in the study, were they compared with others who were not malnourished at baseline? Need to give more details on exclusion and inclusion criteria.
Line 136: What were the routine nutritional practices?
Line 141: How did they instruct participants to use formula milk? Adding to their meals? Need to give more details.
Figure 1: It is not clear that the process diagram is for both groups or just the intervention group. The diagram should clearly show what regimen control or treatment group received/not received at each time point of the study.
Line 176: Malnutrition determination needs a reference.
line 178: distribution of stomach and colon cancer in each group?
Line 191: what was the rationale for choosing paired t-test to test the difference between intervention and control groups?
Line 196: did the authors use matching just in their analysis? Was this study a randomized clinical trial?
Reviewer 2 Report
The manuscript proposed the results of an intervention study in neoplastic patients with stomach and colon cancer, with a control group. The open-label intervention consisted of nutritional counselling, offered to both groups, and the personalization of the nutritional intervention with the aid of nutritional supplements and dietary adaptations based on the guidelines found in the literature. Some outcomes showed significant improvements over a 2-month follow-up on nutritional and body composition indices in the intervention group but not in the control group. The results suggest the importance of nutritional personalization in cancer patients, especially with cancers affecting the digestive system, suffering from malnutrition and consequent worse response to antineoplastic treatments. The manuscript explores aspects that still need knowledge development and a shared approach in clinical practice. Antineoplastic treatments have shown an improvement in the survival of cancer patients over time but such treatments lead to a significant deterioration in the quality of life and often a worse clinical response. In particular, the reference population sample is particularly suitable, taking into account cancer localization such as that of the stomach, which have a high incidence in these countries.
The open-label design, the limited number of participants and the reduced follow-up define the work as a pilot study (quasi-experimental, according to the authors).
Despite the enthusiasm of the authors, at the end of the introduction, I would avoid overinterpreting the results, as the data obtained is not sufficient to define a nutritional guide for the approach to cancer patients.
Enrollment could be better described, specifying the technique with which the participants were invited to the study, how many were proposed, how many accepted and how many, possibly, were excluded because they did not meet the inclusion criteria.
The propensity score-matching allows to reduce the risk of bias in the division between the control group and the treated group, however, it reduces the number of individuals evaluated. Does this lead to underpowered outcomes, based on the authors' sample size calculation?
In the measurement of lean mass and muscle mass, reference is made to bioimpedance analysis. However, this predictive technique is not very effective from a clinical point due to possible discrepancies between the population evaluated and that used for predictive regression formulas. Part of this problem can be overcome by vector bioimpedance analysis. Which bioimpedance instrument and which formulas were used?
On page 7, among the results, there is a discrepancy between some percentage values ​​in the text and those in table 1.
In the discussion, reference is made to the role of physical activity. Did the counselling offered to the participants include this aspect? Why weren't the activity levels of the participants assessed? From the authors' discussion, it seems a relevant aspect.
Reviewer 3 Report
The topic and conclusion drawn is not very original – many studies have shown that nutrition interventions improve nutritional status (regardless of patient group and ethnicity). The manuscript is quite lengthy regarding the amount of words and tables. I suggest it should be majorly shortened and focused. The giant tables are not well readible in my opinion - perhaps the introduction of some figures may help. The description of the methods is not always clear. The English is scientific and appropriate in general, some errors in sentence structure need to be corrected.
As it is now, the manuscript provides little additional knowledge for the (international) general topic “healthcare”.
Some specific issues:
Lines 74-77: error in sentence structure
Line 86: what is meant by quasi-experiment
Line 88: what is a convenience sampling technique?
Line 91: error in sentence structure (either…”or”)
Lines 90-98: so it is to be understood as a case-control study?
Line 98: this is unclear to me - why exclude comorbid patients? These are everyday reality in clinical practice
Line 109: improve English
Chapter 2.3: From the text only, I am unclear as to which intervention was performed at which points of time, perhaps this can be rephrased?
Figure 1: correct grammar “in nutritional counseling”
Lines 157-161: seems overly academic compared to the rather short descriptions of the other measurement instruments
Round 2
Reviewer 1 Report
The quality of this paper has been improved significantly.
Reviewer 2 Report
The authors responded adequately to requests for reviews